# Demonstration of chemistry at a point through restructuring and catalytic activation at anchored nanoparticles

Dragos Neagu [1], Evangelos I. Papaioannou[2], Wan K.W. Ramli[2,3], David N. Miller[1], Billy J. Murdoch[4], Hervé Ménard[5], Ahmed Umar[1], Anders J. Barlow[4], Peter J. Cumpson[4], John T.S. Irvine [1] & Ian S. Metcalfe[2]

Metal nanoparticles prepared by exsolution at the surface of perovskite oxides have been recently shown to enable new dimensions in catalysis and energy conversion and storage technologies owing to their socketed, well-anchored structure. Here we show that contrary to general belief, exsolved particles do not necessarily re-dissolve back into the underlying perovskite upon oxidation. Instead, they may remain pinned to their initial locations, allowing one to subject them to further chemical transformations to alter their composition, structure and functionality dramatically, while preserving their initial spatial arrangement. We refer to this concept as chemistry at a point and illustrate it by tracking individual nanoparticles throughout various chemical transformations. We demonstrate its remarkable practical utility by preparing a nanostructured earth abundant metal catalyst which rivals platinum on a weight basis over hundreds of hours of operation. Our concept enables the design of compositionally diverse confined oxide particles with superior stability and catalytic reactivity.

[1] School of Chemistry, University of St Andrews, St Andrews KY16 9ST, UK. [2] School of Engineering, Newcastle University, Newcastle-upon-Tyne, NE1 7RU, UK. [3] School of Bioprocess Engineering, University Malaysia Perlis, 02600 Perlis, Malaysia. [4] National EPSRC XPS Users' Service (NEXUS), School of Mechanical and Systems Engineering, Newcastle University, Newcastle upon Tyne, NE1 7RU, UK. [5] Sasol (UK) Ltd., St Andrews KY16 9ST, UK. Dragos Neagu and Evangelos I. Papaioannou contributed equally to this work. Correspondence and requests for materials should be addressed to J.T.S.I. (email: jtsi@st-and.ac.uk) or to I.S.M. (email: ian.metcalfe@newcastle.ac.uk)

Recently, the redox exsolution method has emerged as a flexible platform for addressing key challenges facing catalysis and energy conversion and storage technologies in terms of controlling the reactivity and durability of nanoparticles supported on oxide surfaces as well as the effective preparation and deployment of such systems[1–10]. In redox exsolution, metal particles of controlled size are allowed to emerge at the surface of an oxide support under reducing atmosphere[4]. Various metals or metal alloys may be exsolved from a range of host lattice compositions[2–4, 6, 7, 9, 11–15]. Interestingly, exsolved particles are socketed into the oxide surface, seemingly confined and strained, which for base metals seems to unlock levels of functionality otherwise inaccessible through conventional means[5, 15–17]. For example, when prepared by exsolution, Ni nanoparticles are not only more stable against agglomeration but also display greatly improved coking resistance in hydrocarbon catalysis or carbon conversion applications[1, 5, 8, 18]. Additionally, owing to the redox nature of the process, exsolved particles may be produced within minutes electrochemically, enabling quick 'switching on' of high-performance solid oxide fuel or electrolysis cells[19].

However, in its inception, the redox exsolution concept was meant to address particle agglomeration over time by periodically regenerating particles through redox cycling[2]. That is, under oxidising conditions the particles (even if somewhat agglomerated), would oxidise and re-dissolve as ions in the underlying perovskite lattice. When exposed again to reducing conditions, the respective ions would re-emerge and reduce into small, well-dispersed metallic particles at the surface. Although it is widely assumed that regeneration is implicitly related to exsolution, there has been some evidence to suggest this process is not entirely reversible[20, 21]. In particular, an alternative mechanism, where metal particles are oxidised to yield stable oxide particles instead of re-dissolving, has yet to be demonstrated. Clearly such a demonstration could yield oxide nanoparticles with a host of potential applications in catalysis and beyond.

Recent results show that the defect chemistry of the perovskite host lattice may be used to control exsolution[4] and there is reason to believe that defect chemistry would also impact particle re-dissolution upon oxidation. In the original concept, stoichiometric perovskite oxides ($ABO_3$) were used. In this case, the exsolution of $x$ moles of metal from the B-site would be accompanied by $x$ moles of AO oxide formation (see Eq. 1)[2]. Upon oxidation, the AO phase could facilitate particle redissolution by reacting with the oxidised particles to reform the initial perovskite (Eq. 1). However, a perovskite containing $x$ moles of A-site vacancies ($A_{1-x}BO_3$) in principle would be able to exsolve up to $x$ moles of metal from the B-site without releasing AO oxides (Eq. 2). Since under typical conditions only a limited fraction $x$ of the B-site metal exsolve[5], a perovskite with sufficiently high A-site deficiency (e.g. $x \geq 0.1$) is less likely to form AO oxides upon exsolution and more likely to form a stable residual perovskite lattice. Overall, this is expected to decrease the driving force for particle redissolution on oxidation, potentially making exsolution irreversible.

$$\underbrace{ABO_3}_{A-\text{stoichiometric perovskite}} \rightleftharpoons \underbrace{(1-x)ABO_3}_{\text{Residual perovskite}} \\ + \underbrace{xB}_{\text{Exsolved metal}} + \underbrace{xAO}_{\text{Released oxide}} + xO_2 \qquad (1)$$

$$\underbrace{A_{1-x}BO_3}_{A-\text{deficient perovskite}} \rightleftharpoons \underbrace{(1-x)ABO_3}_{\text{Residual perovskite}} + \underbrace{xB}_{\text{Exsolved metal}} + 3/2\,xO_2 \qquad (2)$$

Here we track individual nanoparticles exsolved from highly A-site-deficient perovskites and show they do not re-dissolve upon oxidation but remain fixed in their initial spatial arrangement. Furthermore, taking advantage of the proven coking tolerance of exsolved particles, we explore their behaviour under CO-rich regions of the CO oxidation reaction and observe that depending on particle chemistry, intriguing new nanostructures can be formed. These nanostructures rival a state of the art Pt catalyst for the CO and NO oxidation reactions over hundreds of hours of operation, opening exciting possibilities for the design of base metal oxide particles capable of achieving high site activities in tandem with high stability against agglomeration.

## Results

**Studied systems**. To exemplify the above, we employ two systems, $La_{0.8}Ce_{0.1}Ni_{0.4}Ti_{0.6}O_3$[4], and its Co-substituted variant, $La_{0.7}Ce_{0.1}Co_{0.3}Ni_{0.1}Ti_{0.6}O_3$, in order to enable exsolution of Ni and Co–Ni alloy nanoparticles, respectively (Supplementary Fig. 1). We introduced Co in addition to Ni to extend the range of potential functionalities of the emergent particles. Dense pellets having these compositions were prepared and particles were exsolved at the top surface, to serve as well-defined model catalyst systems for linking particle characteristics (size, shape, population, chemistry etc.) to catalytic and kinetic behaviour (Fig. 1). The catalyst systems were labelled according to the exsolvable elements in the perovskite host lattice, as well as the diameter of exsolved particles prior to any treatment. For example, $CoNi|_P^{30}$ denotes a system with emergent 30 nm Co–Ni-based particles on a perovskite host (P). It should be noted that, whilst the initial emergent particles were metallic (Ni metal or Co–Ni alloy), these evolved into oxidised particles to yield the most active phases throughout the series of experiments discussed herein. The initial microstructure of representative model catalyst samples, $Ni|_P^{30}$, $CoNi|_P^{30}$ and $CoNi|_P^{10}$ is given in Supplementary Figs. 2–4, respectively.

**Towards irreversible exsolution**. Figure 2 links the physical and chemical properties of the model catalysts $Ni|_P^{30}$ and $CoNi|_P^{30}$ to their kinetic behaviour. Figure 2a shows the $CO_2$ production rate from CO oxidation as a function of temperature (henceforth referred to as a light-off experiment). Selected regions of the samples are shown, tracking an average of 20 individual particles before (Fig. 2d, f) and after (Fig. 2e, g) the light-off experiment. These micrographs illustrate that initially the metal particles display a characteristic rounded shape with evidence of slight surface oxidation as revealed by X-ray photoelectron

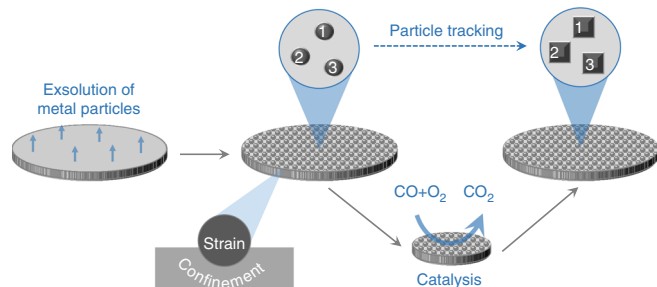

**Fig. 1** Schematic of the particle tracking experiments. Initially, metal particles are exsolved on one side of a dense pellet made from the perovskite composition of choice. Representative areas of this pellet are tagged through geometrical distances to the pellet edge and by proximity to surface features such as pores or terraces (see, for example, Fig. 2f). The desired catalytic test is then carried out, the sample is removed from the reactor and again loaded in the electron microscope. The tagged areas are located and examined to identify the evolution of the particles at individual and collective level and link this to macroscopic catalytic behaviour

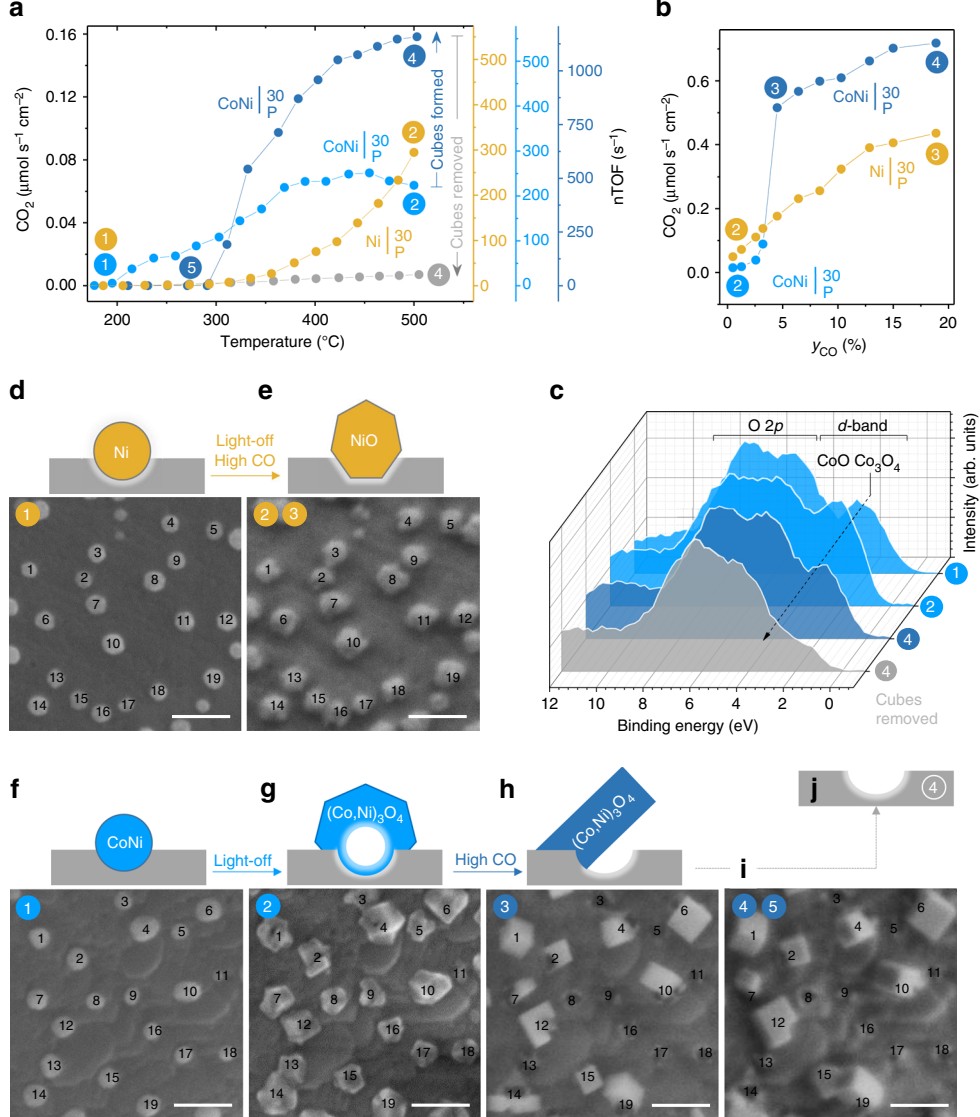

**Fig. 2** Restructuring and catalytic activation at confined particles. Highlighted systems include: $Ni|_P^{30}$ (dark-yellow) and $CoNi|_P^{30}$ (light-blue prior to restructuring and dark-blue after). **a** $CO_2$ production rate from CO oxidation, as a function of temperature, using the indicated model (pellet) catalysts. The rate on the left y-axis is normalised with respect to the area of the pellet surface decorated with particles (Supplementary Note 1). The colour circled numbers highlight the sequence in which the catalytic experiments were carried out and also the stages at which particle tracking or XPS measurements were conducted. The right y-axis shows corresponding nTOF values as a function of temperature (Supplementary Note 3 for details). Errors are of the size of the points used for plotting. **b** $CO_2$ production rate as a function of CO mole fraction, at 520 °C, for the samples used in the experiments shown in **a**. **c** XPS valence band spectra, guideline showing evolution of $Co_3O_4$ phase across selected samples (signal assignment based on ref. [42]). **d–i** SEM micrographs (scale bars, 100 nm) of tracked areas collected at room temperature at the highlighted stages of catalytic testing, and corresponding cross-section schematics of the particle–perovskite interface. **d, e** $Ni|_P^{30}$ microstructure showing: **d** as-prepared metal particles and **e** particles after light-off and CO kinetic experiment. **f–i** $CoNi|_P^{30}$ microstructure showing: **f** as-prepared metal particles; **g** particles after light-off; **h** cubic-like structures formed during the CO kinetic experiment in **b**; **i** final cube microstructure after completion of the CO kinetic (**b**) and an additional light-off experiment shown in **a**. **j** Schematic of the local appearance of the perovskite surface after cube removal (see Supplementary Fig. 5d for SEM micrograph)

spectroscopy (XPS, Fig. 2c, Supplementary Fig. 5). Following the light-off experiment, the particles expand (Supplementary Figs. 6 and 7) and their shape becomes more faceted (Fig. 2e, g), indicating they suffered oxidation. This is consistent with the XPS data which for $CoNi|_P^{30}$ reveals an increase in $Co^{3+}$ at the surface (Fig. 2c).

Notably, both Ni and Co–Ni particles are likely to experience higher strain following this oxidation process (Supplementary Note 1), yet do not re-dissolve into the underlying perovskite host lattice (Fig. 2e, g). Evidence highlighted in Supplementary Note 2 indicates that for the Co–Ni system on average 0.06 B-site atoms

per unit cell exsolved (from up to 200 nm depth), which is much lower than the 0.2 A-site vacancies per unit cell employed in this system. As such, this result is consistent with the premise that an amount of A-site vacancies equal to but preferably greater than the amount of exsolved B-site metal would make redissolution less favourable.

Another notable aspect revealed by particle tracking is that upon oxidation particles do not drift across the perovskite surface as seen in other systems[22]. In fact, the particles maintain the same spatial arrangement as in as-prepared state, suggesting that the well-anchored nature of exsolved particles persists throughout

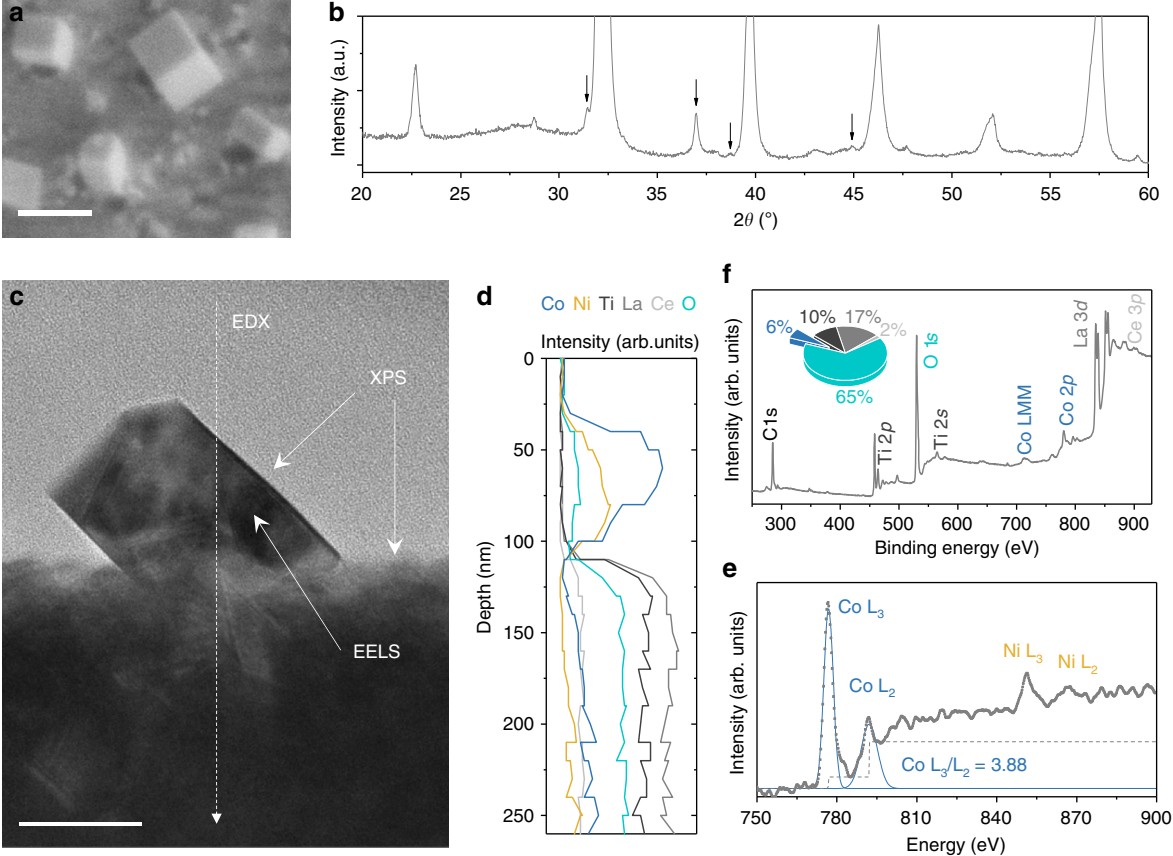

**Fig. 3** The anatomy of activated Co-based systems. **a** SEM micrograph of a representative sample with cubic structures (i.e. activated $CoNi|_P^{30}$); scale bar, 100 nm. **b** XRD pattern corresponding to **a**; the arrows indicate the peaks originating from the spinel structure ascribed to the cubes, while the rest of the major reflections originate from the underlying perovskite. **c** Cross-section TEM micrograph extracted by FIB (Supplementary Fig. 8) from the sample shown in **a**; scale bar, 50 nm. **d** TEM-EDX analysis across the dotted line shown in **c**. **e** EELS spectra from the cube in **c**; the $L_2/L_3$ ratio is indicative of Co +2/+3 mixed oxidation state[43]. **f** Surface analysis and quantification pie chart from XPS corresponding to **a**

oxidation. The observation that the nature of the particles may be altered considerably but their arrangement can be precisely maintained has enabled us to explore further reconstruction at anchored particles, as shown next.

**Chemistry at a point**. Upon increasing the CO mole fraction ($y_{CO}$), the now oxidised $Ni|_P^{30}$ system (Fig. 2e) showed a steady increase in $CO_2$ production rate (Fig. 2b) and preserved particle morphology and arrangement throughout the experiment. Interestingly, in the same conditions, the oxidised $CoNi|_P^{30}$ system (Fig. 2g) displayed one order of magnitude jump in kinetic rates around $y_{CO}$ ~5% (Fig. 2b). Particle tracking revealed that this coincided with a reorganisation of the particles into tilted cubic-like structures (compare Fig. 2g and h). Increasing $y_{CO}$ from ~5% to ~20% only resulted in mild increase in the reaction rates, and brought no further morphological and topological changes to the cubes (Fig. 2i).

To investigate the structure and composition of the cubes and their interface with the perovskite lattice, we employ various techniques as summarised in Fig. 3. Combined evidence by X-ray diffraction (XRD, Fig. 3b), scanning transmission electron microscopy (STEM, Fig. 3c, Supplementary Figs. 8 and 9) and energy dispersive X-ray (EDX, Fig. 3d) indicate that the cubes are generally single-crystal, Co-rich $(Co,Ni)_3O_4$ related spinel structures exposing (100) surfaces. Co is present in mixed oxidation state 2+/3+ both in the bulk and at the surface of the cubes as revealed by electron energy loss spectroscopy (EELS, Fig. 3e) and

XPS (Fig. 2c), respectively. Microstructural and compositional analysis across the cube-perovskite interface (SEM, STEM, EDX in Fig. 3a, c, d, respectively) indicate that cubes are anchored onto socket edges at an angle, forming striking, semi-enclosed loci with the perovskite surface as illustrated schematically in Fig. 2h.

To understand why restructuring into cubes occurred for the Co–Ni oxide particles but not for the Ni-based ones, we examine the different response of these particles to redox conditions in conjunction with their anchored structure. Under light-off (oxidising) conditions, metallic Ni particles undergo a typical oxidation to rock-salt NiO (Fig. 2e). However, the Co in the Co–Ni particles drives oxidation towards a Kirkendall effect, whereby Co atoms diffuse outward[23, 24], forming expanded, hollow-core particles, probably comprising of $(Co,Ni)_3O_4$ related spinel solid solutions (Fig. 2g, Supplementary Fig. 10). Following exposure to higher CO fractions (mildly reducing conditions), the NiO particles appear to remain structurally unchanged, but the hollow-core spinel oxide particles restructure into spinel oxide cubes, a process probably facilitated by their hollow morphology. Notably, however, cubes form and remain adjacent to sockets as if anchored onto them (Figs. 2i and 3a). On one hand, this illustrates the key role played by the initial particle-perovskite interaction in keeping this reorganisation controlled. On the other hand, it is an indication that the innate anchorage of exsolved metal particles may be passed down to their morphed analogues opening interesting possibilities for stabilising base metal oxide nanoparticles. Where outward diffusion and restructuring were more considerable and/or more anisotropic

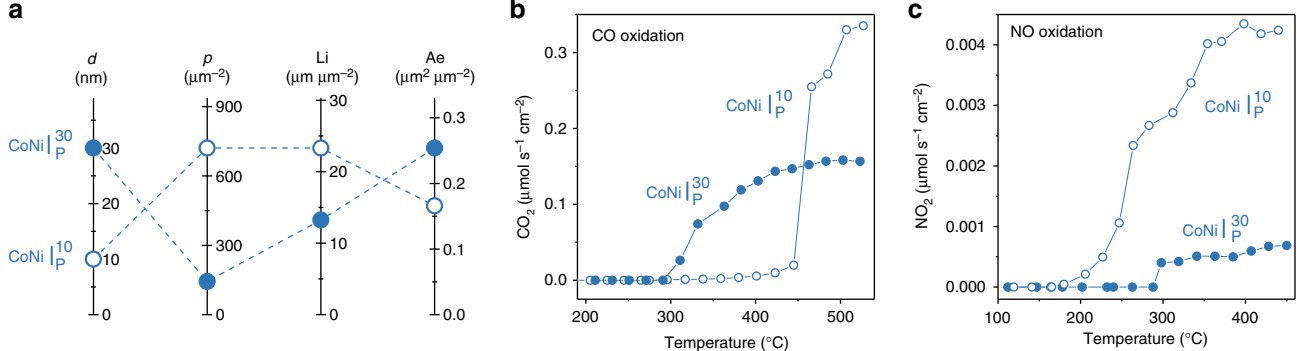

**Fig. 4** The effect of particle characteristics on catalytic activity. **a** Initial particle characteristics for the model pellet catalysts with 10 and 30 nm particles presented as a parallel coordinate plot and including: particle diameter $d$, particle population $p$, length of the particle–perovskite interface Li and exposed particle area, Ae. **b, c** Corresponding catalytic activity after activation, for **b** CO oxidation and **c** NO oxidation. The catalytic rates are normalised with respect to the area of the pellet decorated with particles (Supplementary Note 3). The values of $p$, Li and Ae are normalised with respect to the area decorated with particles (given per $\mu m^2$). The errors are smaller than the points

(Supplementary Fig. 11), adjacent cubes merge (e.g. particles 15 and 19 in Fig. 2g–i), exposing empty sockets in the process (e.g. particles 5 and 8 in Fig. 2g–i). This effectively decreases the particle population, from ~140 initial metal particles per square micron to ~45 spinel oxide cubes per square micron (Supplementary Figs. 2, 7 and 12). A higher fraction of the initial particle population can be maintained throughout the restructuring process if 10 nm size Co–Ni particles are used ($CoNi|_P^{10}$, Supplementary Figs. 4 vs. 13). This is because around or below this particle size the Kirkendall effect is less likely to manifest and the Co–Ni particles undergo a typical oxidation similar to Ni metal particles[25]. At the same time, such small particle size seems to still allow for subsequent restructuring to occur. Overall this enables the possibility to tailor the particles size and, as shown below, also the functionality of microstructures produced through such a chemistry at a point concept.

**Emergent functionality**. While the restructuring of particle arrays through chemistry at a point is fascinating in itself, it actually holds great practical value because it unlocks a whole new level of functionality in these systems. Firstly, upon cube formation in $CoNi|_P^{30}$, catalytic activity in light-off conditions increased nearly 3-fold as compared to the as-prepared state of the system (Fig. 2a). Not surprisingly, no such activation was observed for the Ni-based samples (Supplementary Fig. 14). Secondly, also upon cube formation, the sample with 10 nm particles, $CoNi|_P^{10}$ (Fig. 4a, Supplementary Fig. 13) exhibited 2-fold higher CO oxidation rates at high temperature (Fig. 4b) and 5-fold higher NO oxidation rates across the entire temperature range (Fig. 4c) as compared to the aforementioned $CoNi|_P^{30}$ system. This essentially marks the emergence of additional functionality in this catalyst for a second reaction of practical importance, the NO oxidation reaction[26].

In order to see how the site activities of the initial and restructured particles compare to the literature, we calculate their nominal turnover frequency (nTOF) for the CO oxidation reaction. nTOF represents the number of CO molecules converted to $CO_2$ per second, per exposed metal atom site at the surface of the particles, and was calculated by combining the kinetic and particle tracking data of the model catalysts $Ni|_P^{30}$ and $CoNi|_P^{30}$, as detailed in Supplementary Note 3. It should also be noted that this calculation was enabled by the observation that sample activity is largely given by the presence of the particles. For example, samples without exolved particles or those where the cubes have been removed are virtually inactive (Supplementary Fig. 15, and Fig. 2a, respectively). This is probably not

surprising considering that the surface of the residual perovskite is essentially a Ni/Co-depleted titanate $(La,Ce)TiO_3$, which is expected to exhibit low oxygen mobility and relatively low catalytic activity on its own (see the cross section EDX analysis in Fig. 3d).

The calculated nTOFs values are plotted against temperature on the right axes in Fig. 2a. Surprisingly, even for initial Ni and Co–Ni-based oxide particles the nTOFs are of the order of hundreds per second, that is, a few orders of magnitude higher than generally reported for typical base metal/metal oxide nanoparticles[27]. These values are in the range typically reported for noble metal particles[28], indicating that these base metal oxide particles may have been greatly activated through the strain and/or confinement experienced as a result of their socketed structure. While recent reports have linked strain to enhanced catalytic activity[29–31], to the best of our knowledge, an effect of the magnitude and practical importance described here has yet to be demonstrated for base metal particles.

Not surprisingly, the nTOFs of the cubes are even higher, of the order of thousands per second (dark-blue axis in Fig. 2a), consistent with the observed catalytic activation discussed above. However, both the activation and unusually high nTOFs appear to be counter-intuitive when considering the surface chemistry of the restructured systems. For example, as compared to light-off state, cube formation is marked by an increase in $Co^{2+}$ at the surface (Fig. 2c), which should lead to a decrease in catalytic activity (as opposed to the observed activation), since $Co^{2+}$ is considerably less active than $Co^{3+}$ for the CO oxidation reaction[32]. The increase in $Co^{2+}$ is probably due to the system undergoing overall reduction at higher $y_{CO}$, and the fact that the cubes generally expose (100) faces which are richer in $Co^{2+}$ (Supplementary Fig. 9). Similarly, (100) $Co_3O_4$ nano-cubes are less active than crystals with geometries that expose higher index planes richer in $Co^{3+}$ [33]. Thus, it seems that taken separately, the cubes and the perovskite (essentially inactive if the cubes are removed, Fig. 2a) cannot account for the unusually high activity observed. This suggests that the activity largely originates at the cube-perovskite interface where synergistic effects may occur. This conclusion is also supported by the observation that the $CoNi|_P^{10}$ catalyst, having roughly twice the particle–perovskite interface length (and half the exposed particle area), as compared to $CoNi|_P^{30}$ (Fig. 4a) generally shows better functionality, as discussed above. Overall, this result is consistent with previous reports on the nature of the active sites for the CO oxidation reaction[34]. However, unlike previously, the interfaces developed here through the chemistry at a point concept stand out through

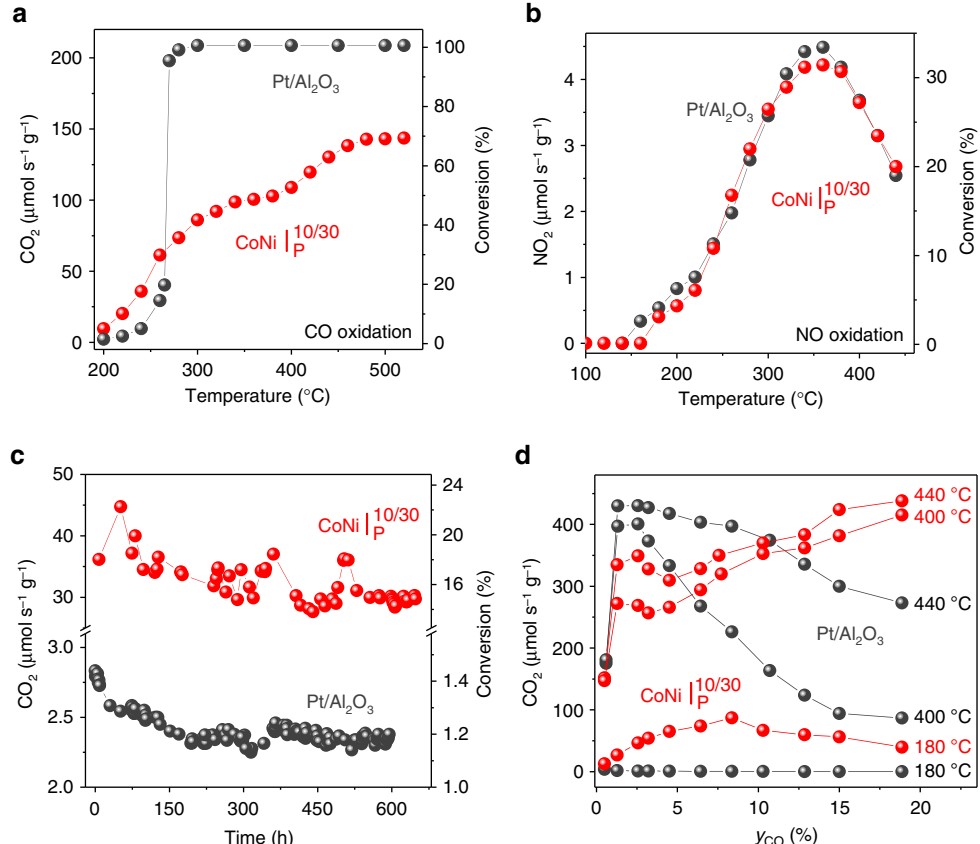

**Fig. 5** Catalytic and long-term performance of powder catalysts. Highlighted powder catalysts are $CoNi|_P^{10/30}$, red and $Pt/Al_2O_3$, black. Catalytic rates are normalised with respect to the catalyst weight (10 mg). **a**, **b** Catalytic rates as a function of temperature for the: **a** CO oxidation reaction and **b** NO oxidation reaction. **c** $CO_2$ production rate as a function of time, over 650 h at 220 °C. Note that in **a** the $Pt/Al_2O_3$ catalyst is operating at full conversion starting around 275 °C. **d** $CO_2$ production rate as a function of CO mole fraction

their semi-enclosed nano-structure and very high nTOFs. This suggests that such interface geometry could greatly enhance activity, providing a new pathway to nanostructure base metal oxide systems to achieve platinum-like functionality.

**A multifunctional catalyst rivalling platinum**. To illustrate the practical value of our findings, we prepared a powder sample, $CoNi|_P^{10/30}$, containing both 10 and 30 nm size metal particles, activated it, and compared it against a state-of-the-art, $Pt/Al_2O_3$ catalyst on a weight-by-weight basis (10 mg). The $Pt/Al_2O_3$ reference catalyst used here is a commercial system from Alfa Aesar having 1 wt.% Pt, 2.7 nm average particle size, 1 $m^2 g^{-1}$ active metal area and exhibits nTOF values characteristic of state-of-the-art noble metal CO oxidation catalysts (see Supplementary Note 3 for values)[35]. The activated $CoNi|_P^{10/30}$ catalyst compares favourably to Pt for CO oxidation, mirroring the combined effect of individual particle sizes (Figs. 4b vs 5a), and, remarkably, matches Pt's activity for NO oxidation across the entire temperature range (Fig. 5b, Supplementary Fig. 16). The outstanding activity for the NO oxidation reaction exhibited by our catalyst is particularly important considering current interest in the mitigation of excess diesel-related NOx emissions[26].

The relatively high activity of $Co_3O_4$ oxide systems for CO oxidation around 250 °C has been long recognised in the literature and ascribed to relatively low barriers for surface reaction and by the fact that $O_2$ adsorption is not inhibited by CO[35]. However, as compared to previous systems, our catalyst comprising of $Co_3O_4$-based cubes anchored at an angle on a perovskite surface, stands out in two key aspects. Firstly, it is able

to maintain high activity over hundreds of hours of continuous operation at temperature (Fig. 5c), in conditions where nanostructured $Co_3O_4$ oxides typically agglomerate and therefore deactivate in just a few tens of hours[35–37]. Secondly, the catalytic rates achieved in our system are considerably higher than those generally observed for $Co_3O_4$-based systems. For example, the highest catalytic rates observed over noble metals at 250 °C lies between 22 and 220 μmol $CO_2$ $s^{-1}$ $m^{-2}$, while for $Co_3O_4$ oxides these are in the range 13–21 μmol $CO_2$ $s^{-1}$ $m^{-2}$ [35]. In similar conditions, we observe rates of ~60 μmol $CO_2$ $s^{-1}$ $g^{-1}$ (Fig. 5a), which, factoring in the total catalyst surface area of 1.5 $m^2 g^{-1}$ (see the experimental section), yields an activity of ~40 μmol $CO_2$ $s^{-1}$ $m^{-2}$ (with respect to the total catalyst surface area). However, the $Co_3O_4$-based cubes in our catalyst only account for ~0.27 of the total catalyst surface area (see the table in Supplementary Fig. 12), meaning that the activity of our catalyst normalised per active area of $Co_3O_4$-based cubes is about ~148 μmol $CO_2$ $s^{-1}$ $m^{-2}$. This value is 7 times higher than the activities cited above for various $Co_3O_4$ oxides and in the upper range of activities observed for noble metals. This result based on catalytic rates is consistent with our conclusion based on nTOFs derived from model (pellet) catalysts, whereby the $Co_3O_4$-like cubic structures produced here through the chemistry at a point concept surpass systems produced by conventional methods by a large margin, approaching activities characteristic to noble metals.

The remarkable catalytic reactivity and stability of these systems complements well their promising applicability. $CoNi|_P^{10/30}$ does not coke and its kinetic behaviour actually improves at higher CO mole fractions, where Pt exhibits declining

reaction rates, due to strong surface CO adsorption which blocks the active sites (Fig. 5d)[38]. Figure 5c shows that catalytic activity is maintained for over 650 h of continuous testing, consistent with the well-anchored nature of these particles, capable of maintaining initial spatial arrangement, as revealed by the tracking experiments. In fact, the tracking experiments exemplify the robustness of these systems. After each catalytic test the samples were removed from the reactor, transported by road to the microscopy facility (170 miles away) repeatedly back and forth to collect the data in Fig. 2, yet the particles not only remained in place but also preserved activity.

Overall, we believe that the concept and results demonstrated above represent a step change in the design of earth-abundant metal catalyst rivalling platinum for reactions of key practical importance such as the CO and NO oxidation, on a weight basis, and also at temperatures of practical significance[39–41]. We demonstrated that redox-tolerant, well-anchored nanoparticles may be produced to serve as platform for further chemical and morphological transformations which may unlock superior functionality in base metal systems. It is also worth noting that aside from enabling the above demonstration, the particle tracking experiments employed here illustrate vividly that the evolution of base metal particles during the CO oxidation reaction and are invaluable for decoding kinetic behaviour and establishing insightful structure-property correlations.

## Methods

**Sample preparation**. Perovskite oxide pellets were prepared by a modified solid state synthesis as described in detail in Supplementary Methods. The as-prepared pellets were further processed for the tracking experiments as explained below, or for the preparation of the powder, they were crushed and ball-milled to produce a powder with a total specific surface area of ~1.5 $m^2 g^{-1}$. For the tracking experiments the pellets were polished on one side to enable uniform exsolution of metal particles. Polishing was carried out with a Metaserv 2000 polisher. Initially, Met-Prep P1200 polishing paper was used, followed by cloth polishing with MetPrep 6, 3 and 1 µm diamond paste, respectively. The samples were cleaned in between each steps in acetone in an ultrasonic bath. To exsolve particles, the samples were reduced in a controlled atmosphere furnace, under continuous flow of 5% $H_2$/Ar (20 ml $min^{-1}$) at the different temperatures with heating and cooling rates of 5 °C $min^{-1}$. The following conditions were used: $CoNi|_P^{30}$ (860 °C, 30 h), $Ni|_P^{30}$ (830 °C, 30 h), $CoNi|_P^{10}$ (550 °C, 30 h). The powder used in catalytic testing, $CoNi|_P^{10/30}$, has a total specific surface area of ~1.5 $m^2 g^{-1}$ and a weight particle loading between 0.8 wt% (10 nm particles) and 3.1 wt.% (30 nm particles).

**Sample characterisation**. The phase purity and crystal structure of the prepared perovskites was confirmed by room temperature XRD by using a PANalytical Empyrean X-ray diffractometer operated in reflection mode. High-resolution secondary and backscattered electron images were obtained using a FEI Scios scanning electron microscope (SEM). This instrument also served for the preparation of a thin lamella by focused ion beam (FIB).

Transmission electron microscopy (TEM), energy dispersive X-ray (EDX) analysis and electron energy loss spectroscopy (EELS) analysis were carried out on a FEI Titan Themis instrument. He-ion microscopy (HIM) was carried out at Newcastle University on a Zeiss ORION NanoFab instrument, using a 25 keV $He^+$ beam with 0.2 pA beam current to image sample $CoNi|_P^{10}$.

X-ray photoelectron spectroscopy (XPS) was carried out at two locations using monochromatic Al X-ray sources. At Newcastle University Thermo Scientific K-Alpha instrument was used while at Sasol, a Kratos Axis Ultra-DLD photoelectron spectrometer. The data was analysed using CasaXPS software. Quantification was performed based on the area of peaks of interest (Ce 3p, La $3d_{5/2}$, Ti 2p, Co 2p) after the subtraction of background of appropriate shape.

**Particle tracking**. The principle used to track areas or particles is described in Fig. 1. The principle used to map particles and calculated their size distribution is described as follows (see refs. [5, 19] also). SEM micrographs were converted to binary images where particles were outlined based on pixel contrast. From this, the number of particles as well as individual particle diameter can be calculated and therefore particles size distribution. Based on this the exposed area and interface length of the particles are calculated as square microns per square microns of pellet area and microns per square microns of pellet area, respectively. The calculations were carried out assuming the particles have hemispheric geometry.

**Catalytic tests**. For the catalytic experiment with the pellets, a continuous-flow single-chamber reactor was used. All experiments were conducted at atmospheric pressure. A K-type thermocouple was used to measure the sample temperature. The gases used were 20% CO/He, 20% $O_2$/He, and CP grade He provided by BOC Ltd. Flow rates of $1 \times 10^{-4}$ mol $s^{-1}$ were used. Helium was used as a balance gas.

To study the effect of temperature, the pellets were heated in an inlet gas mixture of 1% of $O_2$ and 0.6% of CO from 100 °C up to 520 °C. The temperature was held during heating after each step of 20 °C, the holding time being varied depending on the time the reaction rate needed to become steady: i.e., the rate of $CO_2$ production did not vary by more than ±5% over 60 min. To study the influence of CO the $O_2$ inlet mole fraction was held constant at 0.64% and the CO inlet mole fraction was allowed to vary between 0.5 and 18.9%.

For the catalytic experiment with the powders, a fix packed-bed reactor was used. Volumetric dilution within catalyst bed is made by mixing the catalyst powder with $Al_2O_3$ powder (10 wt.% of each catalyst is diluted with $Al_2O_3$ to get a total weight of 100 mg). All experiments were conducted at atmospheric pressure. A K-type thermocouple was used to measure the sample temperature. The gases used were 20% CO/He, 20% $O_2$/He, 1% NO/He and CP grade He provided by BOC Ltd. Flow rates of $3 \times 10^{-4}$ mol $s^{-1}$ were used. Helium was used as a balance gas.

To study the effect of temperature and the influence of CO, similar steps to the experiments with CO with the pellets were followed. In order to be able to measure the rate of $CO_2$ production during the long-term experiment (over 650 h) under gradientless conditions the reactor was operated under conditions of differential conversion.

For the catalytic experiments with NO, to study the effect of temperature, the powders were heated in an inlet gas mixture of 8% of $O_2$ and 0.04% of NO from 100 °C up to 440 °C. The temperature was held during heating after each step of 20 °C, the holding time being varied depending on the time the reaction rate needed to become steady: i.e. the rate of $NO_2$ production did not vary by more than ±5% over 60 min.

Additional details on the setups, conditions and procedures used can be found in Supplementary Methods.

**Data availability**. Data supporting this publication is openly available under an 'Open Data Commons Open Database License'. Additional metadata are available at http://dx.doi.org/10.17630/aa60e158-c7dd-4863-8844-92828a236bfe.

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

## Acknowledgements

The research leading to these results has received funding from the European Research Council under the European Union's Seventh Framework Programme (FP/2007-2013)/ ERC Grant Agreement Number 320725 and from the EPSRC via the research grants EP/ J016454/1, EP/G01244X/1, EP/K015540/1, EP/J018414/1, as well as EPSRC Capital for Great Technologies grants EP/L017008/1 and EP/K022679/1, and a Royal Society Wolfson Merit Award (WRMA 2012/R2). We thank the National EPSRC XPS Users' Service, an EPSRC Mid-Range Facility and Sasol St Andrews for XPS data acquisition.

## Author contributions

Concept: D.N., J.T.S.I., E.I.P. and I.S.M. Methodology: D.N., E.I.P., J.T.S.I. and I.S.M. Experimental: sample preparation, D.N. and A.U.; catalytic testing, E.I.P. and W.K.W.R.; XRD, D.N.; SEM, D.N. and E.I.P.; TEM, D.N.M.; HIM, A.J.B.; XPS, B.J.M. and H.M. Data analysis: D.N. and E.I.P. with assistance from B.J.M. and H.M. for XPS and D.N.M. for TEM. Resources: J.T.S.I., I.S.M. and P.J.C. Manuscript preparation: D.N. with assistance from E.I.P. Supervision J.T.S.I. and I.S.M. All authors discussed the results and commented on the manuscript.

## Additional information

**Competing interests:** The authors declare no competing financial interests.

**Reprints and permission** information is available online at http://npg.nature.com/ reprintsandpermissions/

