## [Peer Review File · Nature Communications]

Editorial Note: This manuscript has been previously reviewed at another journal that is not operating a transparent peer review scheme. This document only contains reviewer comments and rebuttal letters for versions considered at Nature Communications. Mentions of prior referee reports have been redacted.

PEER REVIEW FILE

Reviewers' comments:

Reviewer #1 (Remarks to the Author):

This is an excellent work in which comprehensive techniques were used to characterize the evolution of perovskite oxide with exsolved nanoparticles upon further treatment. The authors clarified the argument within our research community: if in-situ exsolution is reversible. They proved that exsolved particles on a perovskite with sufficiently high A-site deficiency (e.g. $x \geq 0.1$) do not re-dissolve back into the underlying perovskite upon oxidation and may remain pinned to their initial locations. The authors have exemplified that redox tolerant, well-anchored nanoparticles may be produced to serve as the platform for further chemical and morphological transformations which may unlock superior functionality in base metal systems. The new concept “chemistry at a point” that the authors proposed in this manuscript is attractive, and the anchored nature allows the further chemical transformations of the metallic nanoparticles to engineer their compositions, structures and functionalities without changing their initial spatial arrangements. This manuscript presents significant improvements, and the authors provide convincing data to support its conclusions. The manuscript is important to scientists in our field. Therefore, it meets the criteria of Nature Communications. I recommend minor revision. I would like to suggest the authors to address the following point.

It has been reported that the further chemical transformations of the metallic nanoparticles may lead to synergistic catalytic behavior between perovskite host and exsolved metal and/or metal oxide. I am curious about the role of the perovskite in the present study. Does it only serve as a support? It was demonstrated that the oxide was more catalytically active than the metallic nanoparticles. The authors may need to conduct research to show the catalytic performance of the perovskite without particle exsolution to answer this question.

Reviewer #2 (Remarks to the Author):

This is a very interesting work on the reconstruction and reactivity of base metal oxide nanoparticles anchored on perovskite oxides for catalytic CO or NO oxidation reactions, by using SEM, TEM, XPS, XRD and fixed-bed reactor. Different from the dissolution and regeneration of metal nanoparticles into and from perovskite oxides during oxidation and reduction respectively, the authors in this work show Ni and CoNi nanoparticles can maintain the spatial arrangement in oxidation reactions. I find that the studies were well conducted, in

particular the part of same area tracking experiment over model catalytic systems. I recommend the publication in Nature Communications after addressing the following comments.

(1) The authors study the structure and reactivity of Ni and CoNi catalysts with different CO molar fraction. How is the reconstruction of Ni and CoNi nanoparticles in pure O₂ oxidation treatment at different temperatures.

(2) In order to identify the real active sites for catalytic CO and NO oxidation, I recommend the authors also show the catalytic activity as a function of length of the particle-perovskite interface.

(3) What is the reaction temperature in Figure 5(c)?

(4) The authors should discuss and cite the following relevant papers: Appl. Catal. B 2014, 144, 277; Chem. Commun. 2013, 49, 9383

Responses to Reviewers' Comments:

Reviewer #1

This is an excellent work in which comprehensive techniques were used to characterize the evolution of perovskite oxide with exsolved nanoparticles upon further treatment. The authors

clarified the argument within our research community: if in-situ exsolution is reversible.

I recommend minor revision. I would like to suggest the authors to address the following point. It has been reported that the further chemical transformations of the metallic nanoparticles may lead to synergistic catalytic behavior between perovskite host and exsolved metal and/or metal oxide. I am curious about the role of the perovskite in the present study. Does it only serve as a support? It was demonstrated that the oxide was more catalytically active than the metallic nanoparticles. The authors may need to conduct research to show the catalytic performance of the perovskite without particle exsolution to answer this question.

This is certainly an interesting point, we have evaluated the initial perovskite and the metal exsolved perovskite, both separately and within the series of experiments described herein and do not see the same good catalytic performances that are the main focus of the report. See original text page 5, lines 180-184:

"For example, samples without exsolved particles or those where the 'cubes' have been removed are virtually inactive (see Fig. S15, and 2A, respectively). This is probably not surprising considering that the surface of the residual perovskite is essentially a Ni/Co-depleted titanate, (La,Ce)TiO₃, which is expected to exhibit low oxygen mobility and relatively low catalytic activity on its own (see the cross section EDX analysis in Fig 3D). "

We choose not to highlight these points beyond this reporting as we consider that it is very difficult to replicate the surface without performing the exsolution and subsequent redox cycles as the surface composition of the perovskite must be perturbed during these processes and similarly would be changed if the nanoparticles were etched. We have investigated etched surfaces without nanoparticles and see the loss of activity but this is not a definitive test of the surface. Our consideration is that the nanoparticles are essential for activity. These are modified beneficially by

the interaction with the perovskite surface, certainly physically, and quite possibly via chemical interaction also.

Reviewer #2 (Remarks to the Author):

This is a very interesting work on the reconstruction and reactivity of base metal oxide nanoparticles anchored on perovskite oxides for catalytic CO or NO oxidation reactions, by using SEM, TEM, XPS, XRD and fixed-bed reactor. Different from the dissolution and regeneration of metal nanoparticles into and from perovskite oxides during oxidation and reduction respectively, the authors in this work show Ni and CoNi nanoparticles can maintain the spatial arrangement in oxidation reactions. I find that the studies were well conducted, in particular the part of same area tracking experiment over model catalytic systems. I recommend the publication in Nature Communications after addressing the following comments.

(1)The authors study the structure and reactivity of Ni and CoNi catalysts with different CO molar fraction. How is the reconstruction of Ni and CoNi nanoparticles in pure O₂ oxidation treatment at different temperatures.

Yes there are certainly many more directions of chemical control to explore including the temperatures used in evolving the oxides and the oxygen partial pressures. We have already considered a range of conditions; however, there is much scope for the research community to explore in the coming years. For this particular form of nanoparticle, it does seem to require a very delicate variation of redox conditions to achieve these special cubes. We do not see such arrays under more oxidising conditions and we anticipate that O₂ would be too oxidising to form these spinel compositions at the temperatures required to maintain small enough exolutes.

(2)In order to identify the real active sites for catalytic CO and NO oxidation, I recommend the authors also show the catalytic activity as a function of length of the particle-perovskite interface.

We have addressed this in the manuscript

Page 6 This conclusion is also supported by the observation that the $\text{CoNi}|_P^{10}$ catalyst, having roughly twice the particle-perovskite interface length (and half the exposed particle area), as compared to $\text{CoNi}|_P^{30}$ (Fig. 4A) generally shows better functionality

Page 13 **Fig 4. The effect of particle characteristics on catalytic activity.** (A) Initial particle characteristics for the model pellet catalysts with 10 and 30 nm particles presented as a parallel coordinate plot and including: particle diameter d , particle population p , length of the particle-perovskite interface Li and exposed particle area, Ae . (B-C) Corresponding catalytic activity after activation, for (B) CO oxidation and (C) NO oxidation. The catalytic rates are normalized with respect to the area of the pellet decorated with particles (see Supplementary Note 3). The values of p , Li and Ae are normalized with respect to the area decorated with particles (given per μm^2). The errors are smaller than the points.

This required a very considerable effort to be achieved whilst keeping the particles comparable between the two systems. Doing this more extensively would unnecessarily delay publication, whilst we believe this to be a fertile ground for further study by other researchers

(3)What is the reaction temperature in Figure 5(c)?

220°C, this has been added to text, thank you for pointing this out

(4)The authors should discuss and cite the following relevant papers: Appl. Catal. B 2014, 144, 277; Chem. Commun. 2013, 49, 9383

These two papers nicely illustrate the importance of substrate nanoparticle interaction and are an important background argument. Within this manuscript we used a previous report, ie reference 5 to set this context as the importance of the interaction was placed therein. As we have not directly referred to this underlying principle in the current text, we do not think that adding these references is appropriate; however, these will be very helpful in future manuscripts and reviews where we look again at the broad substrate interactions. Many thanks for alerting us to these.

Reviewers' Comments:

Reviewer #2 (Remarks to the Author):

The authors have addressed my comments. The revised MS can now be accepted.